# NUQ: Nonparametric Uncertainty Quantification for Deterministic Neural Networks

## Abstract

This paper proposes a fast and scalable method for uncertainty quantification of machine learning models' predictions. First, we show the principled way to measure the uncertainty of predictions for a classifier based on Nadaraya-Watson's nonparametric estimate of the conditional label distribution. Importantly, the approach allows to disentangle explicitly *aleatoric* and *epistemic* uncertainties. The resulting method works directly in the feature space. However, one can apply it to any neural network by considering an embedding of the data induced by the network. We demonstrate the strong performance of the method in uncertainty estimation tasks on a variety of real-world image datasets, such as MNIST, SVHN, CIFAR-100 and several versions of ImageNet.

## 1 Introduction

It is crucial in many applications of modern machine learning methods to complement the prediction with some sort of a "confidence" score. In particular, deep neural network models, which usually achieve state-of-the-art results in various tasks, are notorious for providing overconfident predictions on data they did not see during training (Nguyen et al., 2015). This issue restricts their wide usage in the fields with high costs of wrong predictions, such as medicine (Miotto et al., 2016), autonomous driving (Levinson et al., 2011; Filos et al., 2020), finance (Brando et al., 2018) and others. Thus, developing a reliable method of quantifying uncertainty is of great interest to researchers and especially practitioners.

The community in recent years made tremendous efforts to develop different uncertainty estimation methods and approaches, including calibration Guo et al. (2017), ensembling (Lakshminarayanan et al., 2017), Bayesian methods (Gal & Ghahramani, 2016), and many others (Ovadia et al., 2019; Wang et al., 2019). Recently, a series of methods of uncertainty estimation based on the single deterministic neural network model was developed (Van Amersfoort et al., 2020; Liu et al., 2020; van Amersfoort et al., 2021). Their primary focus is on ensuring that embeddings of the data obtained on some layer of a network capture the geometrical relationships between the data samples in the input space, which is done via different regularization strategies. Given this property, one can apply a certain approach to capture uncertainty in the embedding space. The crucial property of these methods is the relatively mild required change in architectures and training procedures, which allows the application of the majority of existing deep learning models.

In practice it is usually important to distinguish two types of uncertainty: *aleatoric* and *epistemic* (Der Kiureghian & Ditlevsen, 2009; Kendall & Gal, 2017). The aleatoric uncertainty reflects the internal noise in the data due to class overlap, data markup errors, or other reasons. This type of uncertainty can not be reduced by providing more data. The epistemic uncertainty reflects the model's ignorance of data. We can reduce the uncertainty of this type once we get more data. Epistemic uncertainty, thus, may be used to identify *out-of-distribution OOD data*. If the model can quantify this type of uncertainty, it may abstain from prediction and address it to a human expert. Also, the ability to quantify epistemic uncertainty helps in active learning (Gal et al., 2017), where lack of "knowledge" naturally indicates in which areas we should label samples. Importantly, there is no single definition of uncertainty, and diverse, often heuristic treatments are usually used in practice.

**Summary of the contributions.** In this paper, we develop *a new and theoretically grounded* method of uncertainty quantification applicable to any deterministic neural network model. More specifically, our contributions are as follows.

1. We rigorously define the uncertainty of the model prediction at a particular data point. This is done by direct consideration of the probability of the wrong prediction.

2. We provide corresponding uncertainty estimate by computing the variance of the kernel estimate of conditional density with the appropriately chosen bandwidth.

3. We apply the resulting method of uncertainty estimation in the neural network's embedding space. Our approach complements recent works in uncertainty estimation for deterministic neural networks, which suffered from a lack of a principled method to quantify the uncertainty.

4. We implement the method in a scalable manner, which allows it to be used on large datasets such as ImageNet. The experimental results in misclassification detection and OOD detection tasks show the significant potential of the proposed approach.

The rest of the work is organised as follows. Section 2 describes the theoretical background and the resulting uncertainty estimation method. Section 3 reviews the related works. In Section 4 we describe the experimental evaluation of the proposed method. Section 5 concludes the study.

## 2 NONPARAMETRIC UNCERTAINTY QUANTIFICATION

### 2.1 ESTIMATION UNDER COVARIATE SHIFT

Let's consider the standard binary classification setup $(X, Y) \in \mathbb{R}^d \times \{0, 1\}$ with $(X, Y) \sim \mathbb{P}_{\mathrm{tr}}$. We assume that we observe the dataset $\mathcal{D} = \left\{ (X_i, Y_i) \right\}_{i=1}^n$ of *i.i.d.* points from $\mathbb{P}_{\mathrm{tr}}$.

The classical problem in statistics and machine learning is to find a rule $\hat{g}$ based on the dataset $\mathcal{D}$ which approximates the optimal one:

$$g^* = \arg \min_g \mathbb{P}(g(X) \neq Y).$$

Here $g \colon \mathbb{R}^d \to \{0, 1\}$ is any classifier and the probability of wrong classification $\mathcal{R}_g = \mathbb{P}(g(X) \neq Y)$ is usually called *risk*. The rule $g^*$ is given by the *Bayes optimal classifier*:

$$g^*(x) = \begin{cases} 1, & \eta(x) \geq \frac{1}{2}, \\ 0, & \eta(x) < \frac{1}{2}, \end{cases}$$

where $\eta(x) = p(Y = 1 \mid X = x)$ which is the conditional distribution of $Y$ given $X = x$ under the distribution $\mathbb{P}$.

In this work, we consider a situation when the distribution of the test samples $\mathbb{P}_{\mathrm{test}}$ is different from the one for the training dataset $\mathbb{P}_{\mathrm{tr}}$, i.e. $\mathbb{P}_{\mathrm{test}} \neq \mathbb{P}_{\mathrm{tr}}$. Obviously, the rule $g^*$ obtained for $\mathbb{P} = \mathbb{P}_{\mathrm{tr}}$ might no longer be optimal if the aim is to minimize the error on the test data $\mathbb{P}_{\mathrm{test}}(g(X) \neq Y)$.

In order to formulate a meaningful estimation problem, some additional assumptions are needed. First of all, we assume that the distribution $\mathbb{P}_{\mathrm{test}}$ is unknown at the model construction moment, and only the dataset $\mathcal{D}$ is available. Also, we will assume that the conditional label distribution $p(y \mid x)$ is the same under both $\mathbb{P}_{\mathrm{tr}}$ and $\mathbb{P}_{\mathrm{test}}$. The latter assumption has two important consequences:

1. All the difference between $\mathbb{P}_{\mathrm{tr}}$ and $\mathbb{P}_{\mathrm{test}}$ is due to the difference between marginal distributions of $X$: $p_{\mathrm{train}}(X)$ and $p_{\mathrm{test}}(X)$. The situation when $p_{\mathrm{test}}(X) \neq p_{\mathrm{train}}(X)$ is known as *covariate shift*.

2. The Bayes rule is still valid, i.e., optimal even under $\mathbb{P}_{\mathrm{test}}$.

However, while Bayes rule $g^*$ is still optimal, its approximation $\hat{g}$ might be arbitrary bad under the covariate shift. The reason for that is that we can't expect $\hat{g}$ to approximate $g^*$ well in the areas where we have few samples from the training set or don't have them at all. Thus, some special treatment of covariate shift is required. We will discuss the particular problem statement in the next section.

## 2.2 PROBLEM STATEMENT

We consider a classification rule $\hat{g}(x) = \hat{g}_{\mathcal{D}}(x)$ constructed solely based on the dataset $\mathcal{D}$. Let us start from defining pointwise risk of estimation:

$$\mathcal{R}(x) = \mathbb{P}(\hat{g}(X) \neq Y \mid X = x),$$

where $\mathbb{P}(\hat{g}(X) \neq Y \mid X = x) \equiv \mathbb{P}_{\text{tr}}(\hat{g}(X) \neq Y \mid X = x) \equiv \mathbb{P}_{\text{test}}(\hat{g}(X) \neq Y \mid X = x)$ under the assumptions above. The value $\mathcal{R}(x)$ is independent of covariate distribution $p_{\text{test}}(X)$ and essentially allows to define a meaningful target of estimation which is based solely on the quantities known for the training distribution.

Let us note that the total risk value $\mathcal{R}(x)$ admits the following decomposition:

$$\mathcal{R}(x) = \tilde{\mathcal{R}}(x) + \mathcal{R}^*(x),$$

where $\mathcal{R}^*(x) = \mathbb{P}(g^*(X) \neq Y \mid X = x)$ is Bayes risk and $\tilde{\mathcal{R}}(x) = \mathbb{P}(\hat{g}(X) \neq Y \mid X = x) - \mathbb{P}(g^*(X) \neq Y \mid X = x)$ is an excess risk. Here $\mathcal{R}^*(x)$ corresponds to aleatoric uncertainty as it completely depends on the data distribution. Excess risk $\tilde{\mathcal{R}}(x)$ directly measures imperfectness of the model $\hat{g}$ and thus can be seen as a measure of epistemic uncertainty.

To proceed, we first assume that the classifier $\hat{g}$ has the standard form:

$$\hat{g}(x) = \begin{cases} 1, & \hat{\eta}(x) \geq \frac{1}{2}, \\ 0, & \hat{\eta}(x) < \frac{1}{2}, \end{cases}$$

where $\hat{\eta}(x) = \hat{p}(Y = 1 \mid X = x)$ is an estimate of the conditional density $\eta(x)$.

For such an estimate we can efficiently bound the excess risk via the following classical inequality Devroye et al. (2013):

$$\tilde{\mathcal{R}}(x) = \mathbb{P}(\hat{g}(X) \neq Y \mid X = x) - \mathbb{P}(g^*(X) \neq Y \mid X = x) \leq 2|\hat{\eta}(x) - \eta(x)|.$$

It allows us to obtain an upper bound for the risk:

$$\mathcal{R}(x) \leq \mathcal{L}(x) = \mathcal{R}^*(x) + 2|\hat{\eta}(x) - \eta(x)|,$$

where $\mathcal{R}^*(x) = \min\{\eta(x), 1 - \eta(x)\}$ is just the Bayes risk. While the upper bound still depends on the unknown quantity $\eta(x)$, we will see in the next section that $\mathcal{L}(x)$ allows for an efficient approximation under mild assumptions.

## 2.3 NONPARAMETRIC UNCERTAINTY QUANTIFICATION

### 2.3.1 KERNEL DENSITY ESTIMATE AND ITS ASYMPTOTIC DISTRIBUTION

To execute the approach above we need to consider some particular type of estimator for $\hat{g}$. In this work, we suggest to consider classical kernel-based estimator of the conditional density as it allows for simple description of its asymptotic properties. For an arbitrary class label $c$, the conditional probability estimate can be expressed as:

$$\hat{p}(Y = c \mid X = x) = \frac{\sum_{i=1}^{N} K_h(x_i - x)[y_i = c]}{\sum_{j=1}^{N} K_h(x_j - x)}. \tag{1}$$

Note that in case of $c = 1$ the equation above gives us $\hat{\eta}(x)$.

In our experiments, we consider one dimensional kernel function $K \colon \mathbb{R} \to \mathbb{R}_+$ and construct the resulting kernel in $\mathbb{R}^d$ of the following form:

$$K_h(x - y) = \prod_{j=1}^{d} K\left(\frac{x^j - y^j}{h}\right).$$

Different choices of kernels are possible, see Supplementary Material, Section A.4.1). A well-known fact (see, e.g. Powell (2010)) is that the difference between $\hat{\eta}(x) - \eta(x)$ for properly chosen bandwidth $h$ converges in distribution as follows:

$$\hat{\eta}(x) - \eta(x) \rightarrow \mathcal{N}\left(0, \frac{1}{Nh^d}\frac{\sigma^2(x)}{p(x)}\left\{\int [K(u)]^2 du\right\}^d\right), \tag{2}$$

where $n$ is the number of data points in the training set, $K(\cdot)$ is the kernel used for kernel density estimate (KDE), $h$ is the bandwidth of the kernel; $d$ is the dimensionality of the problem and $\sigma^2(x)$ is the standard deviation of the data label at point $x$.

Note that we can efficiently approximate the variance term in (2). To start with, the integral $\int [K(u)]^2 du$ can be computed in the closed form for various standard kernels, see Supplementary Material, Table 3. Second, we approximate the marginal density of objects $p(x)$. The density can be again obtained via KDE: $\hat{p}(x) = \frac{1}{Nh^d}\sum_{i=1}^N K_h(x - x_i)$. However, one can choose the other estimates of the density, not necessarily related to the considered kernel estimate. For example, Gaussian Mixture Model (GMM) can be used similarly to Mukhoti et al. (2021). The only thing left is the variance which can be estimated as $\hat{\sigma}^2(x) = \hat{\sigma}^2(y \mid x) = \hat{\eta}(x)\left(1 - \hat{\eta}(x)\right)$.

Now we are equipped with an estimate of the distribution for $\hat{\eta}(x) - \eta(x)$. Let us denote by $\tau(x)$ the standard deviation of a Gaussian from equation (2):

$$\tau^2(x) = \frac{1}{Nh^d}\frac{\sigma^2(x)}{p(x)}\left\{\int [K(u)]^2 du\right\}^d.$$

Based on the obtained approximation of the distribution for $\mathcal{L}(x)$, one can construct an abstention procedure that takes into account properties of this distribution and study the theoretical properties of the resulting method similarly to (Zaoui et al., 2020). We defer these studies to future work and focus on the practical approach for obtaining uncertainty estimates which directly follows from the derivations above.

### 2.3.2 ESTIMATES OF TOTAL, ALEATORIC AND EPISTEMIC UNCERTAINTY

In this work, we suggest a particular uncertainty quantification procedure inspired by the derivation above. More specifically, we suggest to consider the following measure of the total uncertainty:

$$\mathbf{U}_t(x) = \min\{\eta(x), 1 - \eta(x)\} + 2\sqrt{\frac{2}{\pi}}\tau(x),$$

which is obtained by considering an asymptotic approximation of

$$\mathbb{E}_{\mathcal{D}}\mathcal{L}(x) = \min\{\eta(x), 1 - \eta(x)\} + 2\mathbb{E}_{\mathcal{D}}|\hat{\eta}(x) - \eta(x)|$$

in a view of (2) and the fact, that $\mathbb{E}|\xi| = \text{std}(\xi)\sqrt{\frac{2}{\pi}}$ for the zero-mean normal variable $\xi$. The resulting estimate upper bounds the average error of estimation at point $x$ and thus indeed can be used as the measure of total uncertainty.

We also can write the corresponding measures of aleatoric and epistemic uncertainties:

$$\mathbf{U}_a(x) = \min\{\eta(x), 1 - \eta(x)\}, \qquad \mathbf{U}_e(x) = 2\sqrt{\frac{2}{\pi}}\tau(x). \tag{3}$$

Finally, the data-driven uncertainty estimates $\hat{\mathbf{U}}_t(x)$ and $\hat{\mathbf{U}}_e(x)$ can be obtained via plug-in using estimates $\hat{\eta}(x)$, $\hat{\sigma}(x)$, $\hat{p}(x)$ and, consequently, $\hat{\tau}^2(x) = \frac{1}{Nh^d}\frac{\hat{\sigma}^2(x)}{\hat{p}(x)}\int \left[K(u)\right]^2 du$.

The generalization of the considered uncertainty measures to the case of multiple classes results in the total uncertainty given by

$$\mathbf{U}_t(x) = \min_c\{1 - \eta_c(x)\} + 2\sqrt{\frac{2}{\pi}}\tau(x),$$

where $\tau^2(x) = \frac{1}{Nh^d} \frac{\max_c\{\sigma_c^2(x)\}}{p(x)} \int [K(u)]^2 du$ and $\sigma_c^2(x) = \eta_c(x)(1 - \eta_c(x))$. The derivation of these formulas can be found in Supplementary Material, Section A.1. We note that the resulting formula for aleatoric uncertainty $\mathbf{U}_a(x) = \min_c\{1 - \eta_c(x)\}$ coincides with classical maximum probability (MaxProb) uncertainty measure.

The only remaining unspecified ingredient of the procedure is the choice of bandwidth $h$ for KDE.

### 2.4 How to Choose Bandwidth Properly?

The choice of the optimal bandwidth parameter is well-developed in the theory of kernel density estimation. For example, one can base on asymptotically optimal values and select the bandwidth accordingly as in Silverman's (Silverman, 2018) or Scott's (Scott, 1979) rules. However, such estimates are usually very crude in practice.

In this work, we consider the choice of bandwidth based on the Improved Sheather–Jones algorithm (Botev et al., 2010). We assume that the bandwidth optimal for the primary problem (density estimation) is also helpful for OOD detection. It is not necessarily so in practice. Thus, it might be beneficial to tune the bandwidth to optimize the quality of OOD detection if some set of OOD points is available at the training time. However, we find that considered estimates perform fairly well in practice, see the experimental evaluation in Section 4.

### 2.5 How to Compute Kernel Estimate when $N$ is Large?

Our nonparametric method involves a sum over the whole available data to compute the estimates. This could be intractable in practice when we are working with large datasets. However, the typical kernel $K_h$ quickly approaches zero with the increase of the norm of the argument: $\|x - x_i\|$. Thus, we can use an approximation of kernel estimates: instead of the sum over all elements in the dataset, we consider the contribution of only several nearest neighbors. It requires a fast algorithm for finding the nearest neighbors. For this purpose, we use the approach of Malkov & Yashunin (2018) based on Hierarchical Navigable Small World graphs (HNSW). It provides a fast, scalable, and easy-to-use solution to the computation of nearest neighbors.

## 3 Related Work

The notion of uncertainty naturally shows up in Bayesian statistics (Gelman et al., 2013), and, thus, Bayesian methods are often used for uncertainty quantification. The idea is to utilize posterior distribution over some hidden variables (or model parameters) to receive an uncertainty estimate. However, exact Bayesian inference is intractable for modern architectures with many parameters, and approximations are used.

Two popular approximation ideas are Markov Chain Monte Carlo sampling (MCMC; Neal et al.) and Variational Inference (VI; Blei et al. (2017)). The former has theoretical guarantees to be asymptotically unbiased, but has high computational cost. A popular alternative is VI-based mean-field approximation with a Gaussian distribution or distributions, enhanced with normalizing flows (Rezende & Mohamed, 2015; Dinh et al., 2017; Papamakarios et al., 2021; Kobyzev et al., 2020). However, standard VI-based methods usually do not scale well enough to apply to really large modern neural networks. That's why some alternatives are considered, such as the Bayesian treatment of Monte-Carlo dropout (Gal & Ghahramani, 2016).

Despite all efforts, these approximations do not reach the state-of-the-art results, which belong to Deep Ensembles (Lakshminarayanan et al., 2017). The ensembles are efficient but again computationally demanding. That's why a series of papers developed ways of approximating the distribution obtained by an ensemble of models by a single probabilistic model (Malinin & Gales, 2018; Malinin et al., 2020; Sensoy et al., 2018). These methods require changing the training procedure and/or at least double the number of trainable parameters.

Recently, another popular type of model for uncertainty quantification was proposed. Specifically, it was proposed to consider a single deterministic neural network model and only apply mild changes to the architecture and training procedure. The crucial idea behind these methods is to ensure that an embedding space induced by the network captures the geometry of the input space. More specifically,

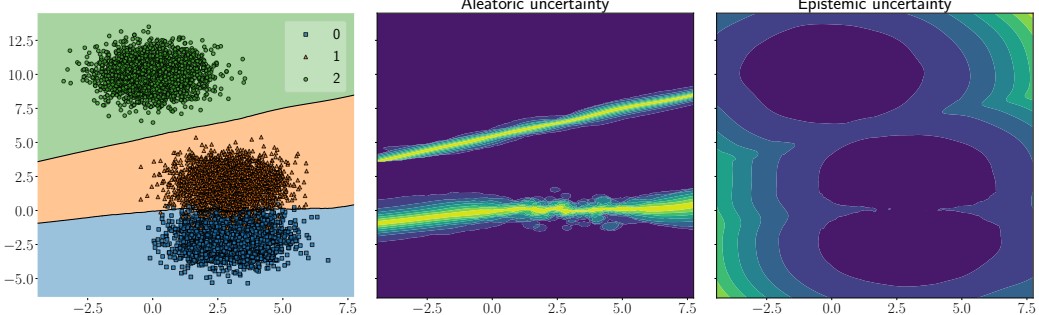

Figure 1: Left plot shows the data and the result of the classification by the Bayes classifier based on the nonparametric estimate of conditional density. The next two plots show different types of uncertainties: "aleatoric" and "epistemic". The lighter color, the higher uncertainty. We see that the former does not increase as we go away from training data, while the latter does.

it was suggested to consider ResNet-type arcitectures (He et al., 2016) with some regularization, which ensures that the model becomes bi-Lipchitz (Van Amersfoort et al., 2020; van Amersfoort et al., 2021; Mukhoti et al., 2021; Liu et al., 2020). These regularization techniques are weight clipping, gradient penalty, and spectral normalization. Deep neural networks trained in this way achieve results comparable with standard approaches but with some training overhead. For uncertainty estimation with such networks variety of approaches were proposed. In DUQ (Van Amersfoort et al., 2020), an RBF layer is added to the network with a custom procedure to adjust the centroid points (in embedding space). The downside of the method is its inability to distinguish aleatoric and epistemic uncertainty. The heuristic to capture epistemic uncertainty was proposed in the DDU approach (Mukhoti et al., 2021) which uses Gaussian Discriminant Analysis in embedding space of a trained neural network for that. SNGP (Liu et al., 2020) and DUE (van Amersfoort et al., 2021) are similar but use a Gaussian process as the final layer, requiring estimating covariance with the use of inducing points or RFF expansion. While aiming for simplicity and minimal changes to the existing approaches for the construction of neural network models, the majority of these methods still require setting additional hyperparameters and/or increased computational cost.

## 4 EXPERIMENTS

### 4.1 TOY EXAMPLE

We start this section with the application of the proposed *Nonparametric Uncertainty Quantification (NUQ)* method to a toy example. As a dataset, we use a 2-dimensional mixture of three Gaussians with centers at points [3, -2], [3, 2], [0, 10], and variance equal to 1. Each Gaussian is treated as a separate class (see Figure 1, the leftmost panel).

We consider the Bayes classifier based on the nonparametric estimate of the conditional density (1) and compute aleatoric and epistemic uncertainty values according to equations (3). Bandwidth was selected according to Improved Sheather–Jones ("ISJ") rule (Botev et al., 2010) independently for each data dimension. Classification results and uncertainties for this toy problem are presented in Figure 1. The first plot shows the raw data and the result of the classification by the Bayes rule. Two other plots present aleatoric and epistemic uncertainty estimates obtained. The uncertainty measures show the desired behavior: aleatoric uncertainty is large in-between the classes, while epistemic uncertainty increases with the increase of the distance to the training data.

### 4.2 IMAGE CLASSIFICATION DATASETS

In this section, we consider a series of experiments on image datasets. In contrast to the toy example above, we should first train a model and then apply NUQ to its predictive features. We emphasise, that NUQ is the postprocessing method, which is fitted to the embeddings obtained from a given model. In what follows, we call this model a "base model". In the experiments of this section, we

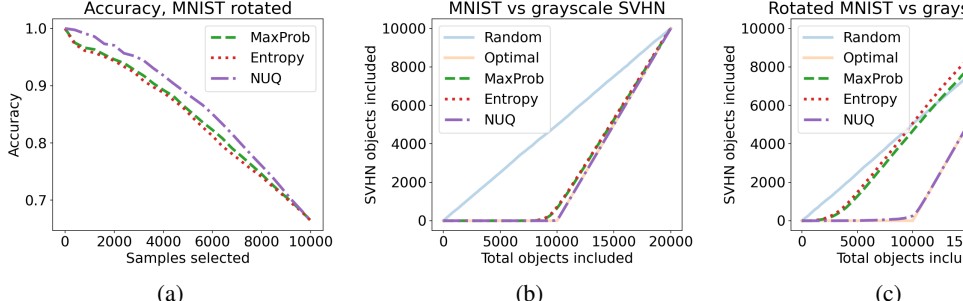

Figure 2: (a) Accuracy for images sorted by uncertainty on rotated MNIST. (b) Share of SVHN images included into consideration vs unrotated MNIST. In this simpler version, even the basic entropy manages to achieve a good result. (c) More challenging task - share of SVHN images included into consideration vs rotated MNIST. NUQ still distinguish between datasets with close to an optimal solution

use logits as extracted features, if not explicitly stated otherwise. However, other options are also possible; see Supplementary Material, Section A.4.

We compare popular measures of uncertainty which do not require significant modifications to model architectures and training procedures. More specifically, we consider:

1. Maximum probability (MaxProb): $1 - \max_c p(y = c \mid x)$;

2. Entropy: $-\sum_{c=1}^{C} p(y = c \mid x) \log p(y = c \mid x)$;

3. Monte-Carlo dropout (Gal & Ghahramani, 2016);

4. Ensemble of models trained with different random seeds;

5. Test-Time Augmentation (TTA) – augmentation, applied to data at inference time;

6. DDU (Mukhoti et al., 2021) involves Gaussian Mixture Model (GMM)-like approximation of extracted features to predict uncertainties.

For Monte-Carlo dropout, Ensembles, and TTA, we first compute average vectors of predictions and then compute its entropy (as we noticed) among MaxProb, Standard deviation, and BALD entropy provides the best ROC-AUC results). More details can be found in Supplementary Material, Section A.2.

### 4.2.1 ROTATED MNIST

The second example is misclassification detection on MNIST (LeCun et al., 2010). We train a small convolutional neural network with three convolution layers, see Supplementary Material, Section A.2. This is the base model we use to obtain logits for the input objects. We consider a particular instance of distribution shift for evaluation by using a test set of MNIST images rotated at a random angle in the range from 30 to 45 degrees. This set contains 10000 images. The range of angles reassures that the data does not look like the original MNIST data, though many resulting pictures can still remind the ones from training.

In this experiment, we consider MaxProb and Entropy-based uncertainty estimates of the base model (using base model predictions, not NUQ) and compare them with NUQ-based estimate of total uncertainty $\hat{\mathbf{U}}_t(x)$. To evaluate the quality of the uncertainty estimates, we sort the objects from the test dataset in order of ascending uncertainties. Then we obtain the model's predictions and plot how accuracy changes with the number of objects taken into consideration; see Figure 2a. The valid uncertainty estimation method is expected to produce the plot with accuracy decreasing when more samples are taken into account. Moreover, the higher is the plot, the better is the quality of the corresponding uncertainty estimate. We see that the plots for all the considered methods show the expected trend, while uncertainties obtained by NUQ are more reliable.

| OOD dataset | MaxProb* | Entropy* | Dropout | Ensemble | TTA | DDU* | NUQ* |
|---|---|---|---|---|---|---|---|
| SVHN | 79.7±1.3 | 81.1±1.6 | 77.6±2.5 | 82.9±0.9 | 81.6±1.2 | **89.6±1.6** | **89.7±1.6** |
| LSUN | 81.5±2.0 | 83.0±2.1 | 76.8±5.1 | 86.5±0.8 | 85.0±2.7 | **92.1±0.6** | **92.3±0.6** |
| Smooth | 76.6±3.5 | 77.8±5.2 | 63.3±3.8 | 83.7±1.2 | 73.2±10.8 | **97.1±3.1** | **96.8±3.8** |

Table 1: OOD detection for CIFAR-100 in-distribution dataset with ResNet-50 neural network. The top two results are shown in bold. Evaluation is done for three models trained with different seeds to estimate the standard deviation. Methods requiring a single pass over the data to compute uncertainty estimates are marked with *.

### 4.2.2 MNIST vs. SVHN

To make the problem more challenging, we consider the SVHN dataset (Netzer et al., 2011), convert it to grayscale, and resize it to the shape of 28 x 28. The size of this additional SVHN-based dataset is again 10000. We take the base model trained on MNIST from the previous section and consider the problem of OOD detection with SVHN being the OOD dataset.

As in-distribution, we first consider the test set of 10000 MNIST images. We again compute uncertainties for each object of this concatenated dataset (10000 of MNIST and 10000 of SVHN) and sort them by their uncertainties in ascending order. For NUQ we use total uncertainty $\hat{\mathbf{U}}_t(x)$ in this experiment. In Figure 2b we plot the share of objects included from the SVHN dataset. It is clearly seen that NUQ assigns higher uncertainties to objects from SVHN. In fact, NUQ almost perfectly separates MNIST from SVHN (optimal result is also depicted on the plot). Although NUQ is the leader in this task, competitors show good performance, and we move on to make the problem more challenging.

We consider the problem of separation between rotated MNIST (see Section 4.2.1) and SVHN. We expect that it is harder to distinguish between them as rotated MNIST images differ from those used to train the network. However, Figure 2c shows that NUQ still does a very good job and allows for almost perfect separation. Interestingly, other methods completely fail and perform no better than random baseline.

### 4.2.3 CIFAR-100

To reinforce our results on simpler datasets, we further conduct experiments on more challenging CIFAR-100 (Krizhevsky, 2009). We want our model to detect the unconventional samples, and thus we treat the out-of-distribution detection as a binary classification task (OOD/not-OOD) by uncertainty score, and we report the ROC-AUC for that task. Following the setup from the recent works (Van Amersfoort et al., 2020; van Amersfoort et al., 2021; Sastry & Oore, 2020), we use SVHN, LSUN (Yu et al., 2015) and Smooth (Hein et al., 2019) datasets as OOD datasets.

We trained the ResNet-50 model from scratch on CIFAR-100. For our method and DDU, we use training with spectral normalization (Miyato et al., 2018) to ensure the bi-Lipschitz constraint for mappings at each layer. In this experiment, NUQ was applied to the features from the penultimate layer, and the density estimate is given by GMM. See the results for other choices of hyperparameters in the Supplementary Material, Section A.4.

The results are presented in Table 1. The ensemble has a strong performance, which is expected. The TTA performs reasonably well with the quality close to the one of the ensemble. We can clearly see that NUQ and DDU show close results while outperforming the competitors with a significant margin.

One may ask whether nonparametric classification method used in NUQ, trained on some embedding from the base model, has any relation to the original neural network. To reassure the reader, we provide an argument that it well approximates the predictions of the base model and NUQ-based uncertainty estimates can be used for the base model as well. Specifically, we compute the agreement between predictions obtained from the Bayes classifier based on kernel estimate (i.e. the one used in NUQ) and base models' predictions. This metric formally can be defined as agreement$(\hat{p}, p) = \frac{1}{n} \sum_{i=1}^{n} I \left[ \arg \max_j \hat{p}(y = j \mid x_i) = \arg \max_j p(y = j \mid x_i) \right]$. For CIFAR-100, this metric gives us the agreement of 0.975, which tells that the approach is accurate.

| OOD dataset | MaxProb* | Entropy* | TTA | Ensemble | DDU* | NUQ* |
|---|---|---|---|---|---|---|
| ImageNet-R | 80.4 | 83.6 | 85.8 | 84.4 | 80.1 | **99.5** |
| ImageNet-O | 28.2 | 29.1 | 30.5 | 51.9 | 74.1 | **82.4** |

Table 2: ROC-AUC score for ImageNet out-of-distribution detection tasks for different methods. Methods requiring a single pass over the data to compute uncertainty estimates are marked with *.

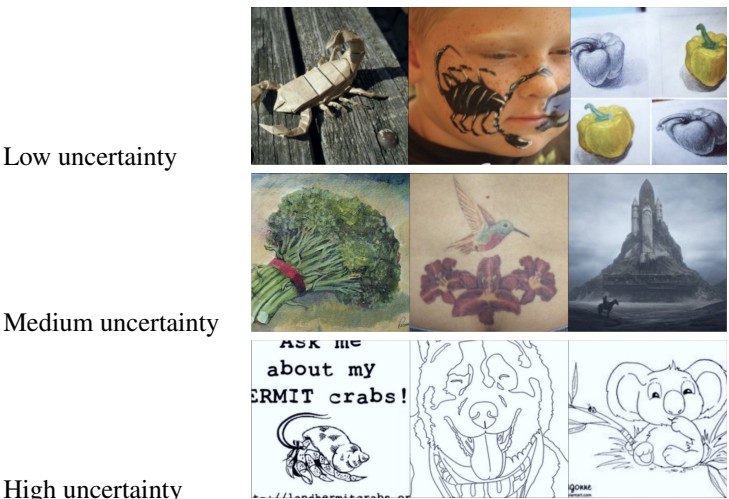

Low uncertainty

Medium uncertainty

High uncertainty

Figure 3: Typical OOD images for different levels of uncertainty as predicted by NUQ.

### 4.2.4 IMAGENET

To evaluate the method's applicability to the large-scale data, we have applied our approach to the ImageNet (Deng et al., 2009) dataset. As OOD data we used the ImageNet-O (Hendrycks et al., 2021b) and ImageNet-R(Hendrycks et al., 2021a) datasets. ImageNet-O consists of images from classes that are not found in the standard ImageNet-1k dataset. ImageNet-R contains different artistic renditions of ImageNet classes.

It turned out that in these experiments, NUQ beats all the competitors with a large margin; see Table 2. The advantage over DDU can be explained by the fact that DDU involves GMM-like approximation of the feature density, one mixture component per class. In ImageNet, there are 1000 classes, and it appears to be an issue for GMM to approximate in such a high-dimensional space efficiently. Also importantly, the resulting inference time for the NUQ estimator was quite fast, with less than 5 minutes needed to process the whole dataset on a moderate machine.

Additionally, we looked at some typical samples from these datasets with low, moderate, and high levels of uncertainty as assigned by NUQ, see Figure 3. We observe that the uncertainty value corresponds well to the intuitive degree of complexity for these images compared to the original ImageNet data. Interestingly, the images with a low degree of uncertainty look like those that can be easily classified by a typical neural network trained on ImageNet.

## 5 CONCLUSIONS

In this work, we propose NUQ, a new principled uncertainty estimation method that applies to a wide range of neural network models. It does not require retraining the model and acts as a postprocessing step working in the embedding space induced by the neural network. NUQ significantly outperforms the competing approaches with only recently proposed DDU method (Mukhoti et al., 2021) showing comparable results. Importantly, in the most practical example of OOD detection for ImageNet data, NUQ shows the best results with a significant margin. All the code to reproduce the experiments is available at http://github.com/omitted/to/preserve/anonymity.

We hope that our work opens a new perspective on model-free uncertainty quantification methods for deterministic neural networks. Since the usage of deep models in high-risk real-world applications is usually limited due to the inability to account for uncertainty, we believe that our ideas could help the community bridge this gap. We also believe that NUQ is suitable for in-depth theoretical investigation, which we defer to future work.

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

# A   SUPPLEMENTARY MATERIAL

## A.1   MULTICLASS GENERALIZATION FOR UNCERTAINTIES

In this section we show, how our method can be generalized from binary classification to multiclass problems. Consider data pairs $(X, Y) \sim \mathbb{P}$. Now, $X \in \mathbb{R}^d$ and $Y \in 1, \ldots, C$, where $C$ is the number of classes. We also denote $\eta_c(x) = \mathbb{P}(Y = c \mid X = x)$.

Let us start with the Bayes risk:

$$\mathbb{P}(Y \neq g^*(X) \mid X = x) = 1 - \mathbb{P}(Y = g^*(X) \mid X = x)$$
$$= 1 - \max_c \eta_c(x) = \min_c \big\{1 - \eta_c(x)\big\},$$

where $g^*(x) := \arg\max_c \eta_c(x)$ is Bayes optimal classifier.

Let us further move to the excess risk. Denote by $\hat{\eta}_c(x)$ density, we approximate. Analogously, $g(x) := \arg\max_c \hat{\eta}_c(x)$

$$\mathbb{P}(Y \neq g(X) \mid X = x) - \mathbb{P}(Y \neq g^*(X) \mid X = x) = \eta_{g^*(x)}(x) - \eta_{g(x)}(x)$$
$$= \eta_{g^*(x)}(x) - \hat{\eta}_{g^*(x)}(x) + \hat{\eta}_{g^*(x)}(x) - \hat{\eta}_{g(x)}(x) + \hat{\eta}_{g(x)}(x) - \eta_{g(x)}(x)$$
$$\leq \big|\eta_{g^*(x)}(x) - \hat{\eta}_{g^*(x)}(x)\big| + \big|\eta_{g(x)}(x) - \hat{\eta}_{g(x)}(x)\big|,$$

where we used the fact that $\hat{\eta}_{g^*(x)}(x) - \hat{\eta}_{g(x)}(x) \leq 0$ for any $x$.

The expectation of the right hand can be upper bounded by $2\sqrt{\frac{2}{\pi}}\tau(x)$, where $\tau(x)$ is defined below.

Total uncertainty for multiclass problem is thus

$$\mathbf{U}_t(x) = \min_c \big\{1 - \eta_c(x)\big\} + 2\sqrt{\frac{2}{\pi}}\tau(x),$$

where

$$\tau^2(x) = \frac{1}{Nh^d} \frac{\max_c \big\{\sigma_c^2(x)\big\}}{p(x)} \int \big[K(u)\big]^2 du$$

and $\sigma_c^2(x) = \eta_c(x)\big(1 - \eta_c(x)\big)$.

## A.2   ARCHITECTURES

### A.2.1   BASE MODEL

For CIFAR-100 and ImageNet-like datasets, we are using ResNet50 with or without spectral normalization (Miyato et al., 2018). For the spectral normalization, we use 3 iterations of the power method. We use a ResNet50 architecture with implementation from PyTorch (Paszke et al., 2019). This architecture was implemented for the ImageNet dataset; thus, for the CIFAR-100, we had to adapt it. We changed the first convolutional layer and used kernel size 3x3 with stride 1 and padding 1 (instead of kernel size 7x7 with stride 2 and padding 3). For CIFAR-100, we train the model for 200 epochs with an SGD optimizer, starting with a learning rate of 0.1 and decaying it 5 times on 60, 120, and 160 epoch. For ImageNet, we train the model for 90 epochs with an SGD optimizer learning rate decaying 10 times every 30 epochs.

For MNIST, we train a small convolutional neural network with three convolution layers with padding of 1 and kernel size of 3. Each of these layers is followed by a batch normalization layer. Finally, it has a linear layer with Softmax activation. This network achieves an accuracy of 0.99 on the holdout set.

We refer readers to our code for more specific details.

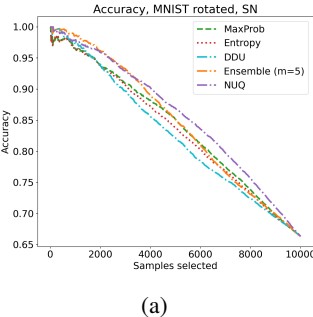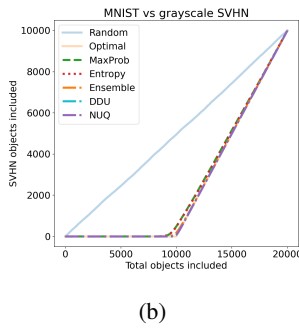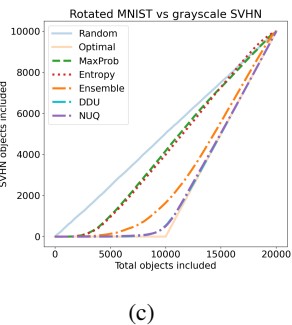

(a)             (b)             (c)

Figure 4: (a) Accuracy for images sorted by uncertainty on rotated MNIST. (b) Share of SVHN images included into consideration vs unrotated MNIST. In this simpler version, even the basic entropy manages to achieve a good result. (c) More challenging task - share of SVHN images included into consideration vs rotated MNIST. NUQ still distinguish between datasets with close to an optimal solution

### A.2.2 Ensemble

For ensemble with use a combination of 5 base models, trained with different random seeds.

### A.2.3 Test-Time Augmentation (TTA)

For TTA, we use a base model with applying a transformation on the inference stage. Images of CIFAR-100 are randomly cropped with padding 4, randomly horizontally flipped, and randomly rotated up to 15 degrees. ImageNet is randomly cropped from 256 to 224, randomly horizontally flipped, and the color was jittered (0.02).

### A.2.4 Spectrally Normalized Models

For both DDU and NUQ, we need spectral normalized models to extract features. We're wrapping each convolutional and linear layer with spectral normalization (PyTorch implementation). We used 3 iterations of the power method in our experiments.

### A.3 More baselines on rotated MNIST benchmark

We provide two more baselines for the rotated MNIST experiments: Deep ensemble and DDU. Overall, we consider this example being too simplistic to make general conclusions and the experiment is included in the main paper is mostly for illustrative purposed and sanity check. Nevertheless, more complete picture with additional based is presented on Figure 4. The NUQ distinctly outperforms ensembles on all examples and is on-par with DDU on out-of-distribution test. However, NUQ is better then DDU in the example with rotated MNIST, see Figure 4a. The possible reason is that NUQ considers total uncertainty which is natural in this examples as rotated-MNIST has naturally the same classes as MNIST and we are in misclassification detection scenario (not pure OOD detection). DDU is worse as it targets only epistemic uncertainty.

### A.4 Ablation Study on CIFAR-100

### A.4.1 Choice of Kernel for Uncertainty Quantification

In this section, we study the choice of a kernel for uncertainty quantification.

We consider the following choices:

We need a probability density estimation for our method, and there are different options: we consider kernel method with RBF kernel and logistic kernel and Gaussian mixtures models. There is also a question about which embeddings to use - the DDU paper proposes to take the features from the second last layer; we believe the logits from the last layer are a reasonable choice as well. To validate

| Kernel name | Formula $K(u)$ | Integral $\int K(u)^2 du$ |
|---|---|---|
| Gaussian (RBF) | $\frac{1}{\sqrt{2\pi}}\exp\left\{-u^2\right\}$ | $\frac{1}{2\sqrt{\pi}}$ |
| Sigmoid | $\frac{2}{\pi}\frac{1}{\exp\{-u\}+\exp\{u\}}$ | $\frac{2}{\pi^2}$ |
| Logistic | $\frac{1}{\exp\{-u\}+2+\exp\{u\}}$ | $\frac{1}{6}$ |

Table 3: Different types of kernels $K(u)$ considered and corresponding values of the integral $\int K(u)^2 du$.

the options, we conducted some ablation study on out-of-distribution detection for the CIFAR-100 dataset, similar to the main experiment.

First, we compare the DDU and NUQ on embeddings from the pre-last and last layer (Table 4) on SVHN, LSUN, and Smooth datasets. Secondly, we compare the NUQ method on RBF, logistic kernel, and GMM for both last and penultimate layer embeddings(Table 5). As we can see from the tables, the optimal is the option with GMM density on the penultimate layer.

| | DDU, features | DDU, logits | NUQ, features | NUQ, logits |
|---|---|---|---|---|
| SVHN | 89.6±1.6 | 88.2±0.6 | 89.7±1.6 | 88.2±0.6 |
| LSUN | 92.1±0.6 | 90.9±0.4 | 92.3±0.6 | 90.9±0.4 |
| Smooth | 97.1±3.1 | 96.3±4.1 | 96.8±3.8 | 96.2±4.1 |

Table 4: Comparison of DDU and NUQ predictions on different type of embeddings - logits (last layer) and features (second last layer).

| | RBF, f | RBF, l | Logistic, f | Logistic, l | GMM, f | GMM, l |
|---|---|---|---|---|---|---|
| SVHN | 84.4±3.2 | 84.7±3.1 | 84.8±2.9 | 86.7±2.6 | 89.7±1.6 | 88.2±0.6 |
| LSUN | 88.2±1.0 | 88.1±0.8 | 88.5±4.0 | 90.3±1.0 | 92.3±0.6 | 90.9±0.4 |
| Smooth | 85.5±6.8 | 87.7±9.4 | 86.2±8.2 | 90.8±7.8 | 96.8±3.8 | 96.2±4.1 |

Table 5: Probability density methods comparison – radial basis function kernel (RBF), logistic kernel, gaussian mixture models (GMM). 'f' (Features) marks models, built on embeddings from a second last layer and 'l' (logits) is for the ones built on embeddings from a last layer.

Kernel-based methods rely on the "reasonable" geometry of the embedding space, meaning that embeddings of similar images should not be too far and different images should not collapse into a single point. Our motivation to use spectral normalization during training is to make the embedding space more smooth with respect to input images. We have conducted an extra ablation study, comparing the result for feature extractors with and without spectral normalization, see Table 6. The results confirm our hypothesis, as the spectral-normalized version performs better, though the NUQ beats the baseline even without applying the modification to the ResNet training. We also show here that entropy performs better than maximum probability as an uncertainty measure.

| OOD dataset | MaxProb | Entropy | DDU | DDU (spectral) | NUQ | NUQ (spectral) |
|---|---|---|---|---|---|---|
| SVHN | 79.7±1.3 | 81.1±1.6 | 88.7±4.3 | 89.6±1.6 | 86.8±1.2 | 89.7±1.6 |
| LSUN | 81.5±2.0 | 83.0±2.1 | 91.3±0.9 | 92.1±0.6 | 91.2±1.1 | 92.3±0.6 |
| Smooth | 76.6±3.5 | 77.8±5.2 | 95.7±1.2 | 97.1±3.1 | 95.5±1.3 | 96.8±3.8 |

Table 6: Comparing the influence of spectral normalization on the model performance for OOD detection, ROC-AUC.

## A.5 PERFORMANCE DIFFERENCE ON CIFAR-100 AND IMAGENET

One of the things that caught our attention is superior performance of NUQ on ImageNet, given that it has very similar results with DDU on CIFAR-100. One of our hypotheses was that embeddings have more complex and multi-modal distribution for more complex Imagenet dataset compared to simpler CIFAR-100. To check this, we made t-SNE based embeddings of out-of-distribution and test

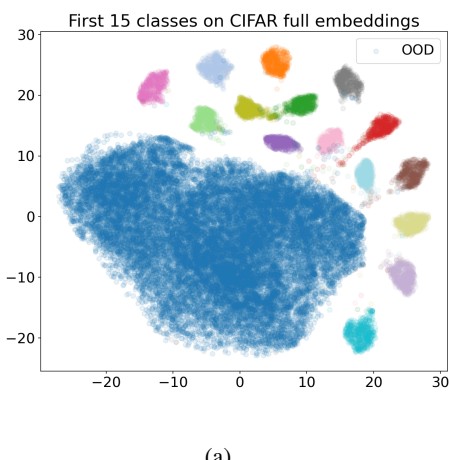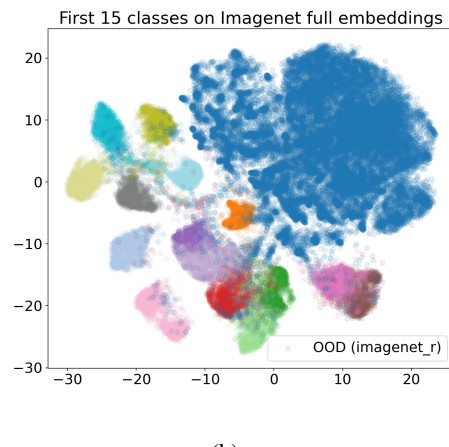

(a)  (b)

Figure 5: Embeddings space visualization for CIFAR (a) and ImageNet (b). We present the embeddings for first 15 classes on test dataset (in various colors) and all the embeddings for out-of-distribution datasets (in blue). The OOD dataset for CIFAR is SVHN and for ImageNet it is ImageNet-R.

data (see Figure 5). While we understand the limitation of this type of visualization, the ImageNet embeddings appear to be much more irregular compared to well-shaped clusters for CIFAR-100. Because of that the modelling of the class with single Gaussian (in DDU) might not work very well for ImageNet. NUQ approach performs the modelling of distributions in much more flexible way which is beneficial for approximation of complex distributions. We hypothesize that this is the reason of the NUQ's superior performance.

### A.6 Toy experiment on detecting actual aleatoric and epistemic uncertainties

In this section, we conduct a toy experiment, for which we explicitly know what should be the true probability of class one, as well as the true data density.

Let us consider a binary classification problem. Our dataset consists of 5000 samples from three different one-dimensional Gaussians, located so that classes are mixed (see picture). Colors denote class label: red - 0; green - 1 (Left in Figure 6) For this particular data model, we can compute the conditional probability of a data point x belongs to class 1: $p(y = 1|x)$. We build an estimate of this conditional using our Nadaraya-Watson kernel-based approach. Further, we generate a uniform grid, and for each point of this grid, using our method, we can upper bound difference between the true conditional and our approximation. This difference, according to our approach, is considered as an epistemic uncertainty (Middle in the Figure 6). The green line in this plot denotes an absolute difference between the true conditional and our approximation. The red line denotes our epistemic uncertainty. From the picture, we can see that our epistemic uncertainty approximates the probabilities difference well. Next, we show how our aleatoric uncertainty relates to the true class 1 conditional probability. In the right plot of the Figure 6 we show true conditional distribution $p(y = 1|x)$ and our approximation of the aleatoric uncertainty. We can see that our approximation is high exactly in the same regions where the true conditional is absolutely unsure about the class label.

### A.7 Pseudo-code

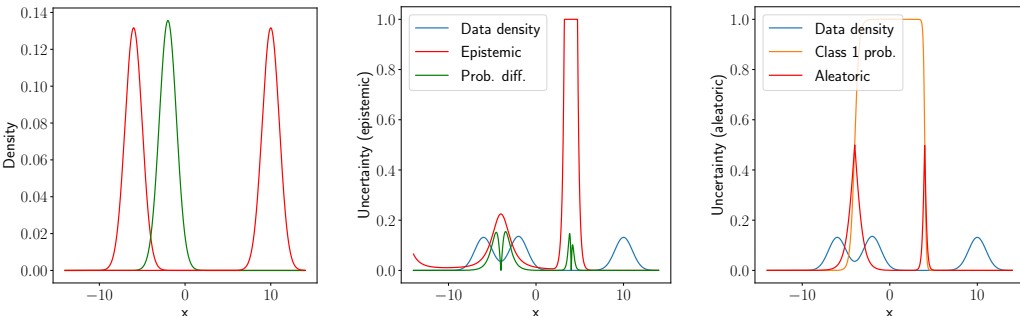

Figure 6: Left: Mixture of one dimensional Gaussians we took samples from. Color denotes class label. Middle: Epistemic uncertainty our model assigns to data points. Note that the uncertainty is quite high in the region of 3-5. For the sake of visualization, we clipped the maximum value to be 1. Right: Our approximation of aleatoric uncertainty is built along with the true conditional probability.

**Input:** Training set $\{(\boldsymbol{x}_i, y_i)\}_{i=1}^N$, inference point $\boldsymbol{z}$
**Output:** Prediction $y(\boldsymbol{z})$ and uncertainty $u(\boldsymbol{z})$

$\{\boldsymbol{x}_{i_k}\}_{k=1}^K \leftarrow K$ nearest neighbours of $\boldsymbol{z}$ among $\{\boldsymbol{x}_i\}_{i=1}^N$

$\eta_c(\boldsymbol{z}) \leftarrow \dfrac{\sum_{k=1}^K K_h(\boldsymbol{x}_{i_k} - \boldsymbol{z})[y_{i_k} = c]}{\sum_{k=1}^K K_h(\boldsymbol{x}_{i_k} - \boldsymbol{z})}$

$\sigma_c^2(\boldsymbol{z}) = \eta_c(\boldsymbol{z})\big(1 - \eta_c(\boldsymbol{z})\big)$

$y(\boldsymbol{z}) \leftarrow \underset{c}{\mathrm{argmax}}\ \eta_c(\boldsymbol{z})$

$p(\boldsymbol{z}) \leftarrow \dfrac{1}{Nh^d} \sum_{k=1}^K K_h(\boldsymbol{x}_{i_k} - \boldsymbol{z})$

$\tau^2(\boldsymbol{z}) \leftarrow \dfrac{1}{Nh^d} \dfrac{\max\limits_c \sigma_c^2(\boldsymbol{z})}{p(\boldsymbol{z})} \int \big[K(u)\big]^2 du$

$u(\boldsymbol{z}) \leftarrow \min\limits_c \{1 - \eta_c(\boldsymbol{z})\} + 2\sqrt{\tfrac{2}{\pi}}\tau(\boldsymbol{z})$

**Algorithm 1:** NUQ inference algorithm.

