# OpenReview forum: "NUQ: Nonparametric Uncertainty Quantification for Deterministic Neural Networks"
_ICLR.cc/2022/Conference — ICLR 2022 Submitted_

### Official Review · Reviewer_CWxx · 2021-11-01

**Correctness:** 3
**Technical Novelty And Significance:** 2
**Empirical Novelty And Significance:** 2
**Recommendation:** 5
**Confidence:** 4

**Main Review:**

Using the logits as input to the kernel machinery may be problematic, as highlighted for example by Postels et al 2021 (arXiv:2107.00649), because these logits will be invariant to many directions in input space (especially if the input is high-dimensional, like an image, compared to the number of classes). It means that the output logits may be invariant to a large volume of changes that may occur when going from in-distribution to out-of-distribution (in fact to volume of output-sensitive directions may be of null measure), making ANY method unable to detect changes in those directions. Hence I would advise against using the output logits.

It is not clear to what extent NUQ is fundamentally different from DUQ (yes, using a sum of kernels rather than a mixture of Gaussians, depending on how many Gaussians are fitted, of course) and the results seem to be very similar between the two in table 1.

For MNIST vs greyscale SVHN, I see in fig 4 that NUQ does the same as the stupid entropy, which is interesting to observe and suggests that in this case it is easy to detect these OOD examples. I also notice that none of the more 'serious' competitive baselines (like Ensembles and DDU) were included in the evaluation. Why?

How were the other methods optimized? have you optimized the number of networks in the Ensemble, for example? How about MC-dropout: how are its hyper-parameters optimized? It does not look like the authors did a serious job of optimizing the hyper-parameters of the other methods.

How accurate is the proposed method in terms of estimating the TRUE epistemic uncertainty? In cases where the generative distribution is known, this should be easy to compute. And how would that compare with PROPERLY OPTIMIZED alternatives?


**Summary Of The Paper:**


This paper proposes NUQ, a kernel-based estimation of epistemic uncertainty regarding a kernel-based probabilistic classifier applied on the input or on the top hidden layer on on output logits of a pre-trained deep net classifier.

Experiments on OOD vs in-distribution binary classification based on the uncertainty score suggest that NUQ works better than several other methods.


**Summary Of The Review:**


The proposed approach is suspicious on theoretical grounds, in that it relies essentially on the ratio of some estimation of training uncertainty and density in representation-space in order to estimate epistemic uncertainty.

The experimental results are also worrisome because it does not look like appropriate efforts were made to tune the hyper-parameters of baseline methods.

Finally, there is no attempt to actually compare the estimated epistemic uncertainty with the ground truth epistemic uncertainty.

---

> ### Author Response · Authors · 2021-11-23
> **Important remarks**
>
> Thank you for your review! Please, see below the answers to your questions and concerns. Please, note that we didn’t perform any major updates to the main text and the answers for your particular concerns were put to Appendix of the paper. We do it for convenience of the review process and will update the main text accordingly if the paper is accepted.
>
> "Using the logits as input to the kernel machinery may be problematic, as highlighted for example by Postels et al 2021 (arXiv:2107.00649)"
>
> The mentioned paper is indeed interesting, and we did not see it before your comment. It is interesting to investigate the effect mentioned in the paper, especially considering how good deterministic neural nets work in uncertainty prediction. However, we should note that penultimate level features can still have directions close to invariance considering the complicated multilayer mappings we have in modern NNs.That’s why we think that the ultimate choice of feature representation should depend on the actual performance of the method. Experimentally we found that logits are also working very good, see our experiments in Appendix A4.1.
>
>
> --------------
>
> "Not clear to what extent NUQ is fundamentally different from DUQ (yes, using a sum of kernels rather than a mixture of Gaussians, depending on how many Gaussians are fitted, of course) and the results seem to be very similar between the two in table 1."
>
>
> Fundamentally, you either fit a Gaussian for class or have a very flexible nonparametric model. As we show in Appendix A.5 the embedding structure for ImageNet seems to be much more complicated than just Gaussians. The other point is that we do not require the retraining of base network as in DUQ. Finally, our method is sound from statistical point of view and we expect to coplement the study with statistical analysis in future works.
>
>
> --------------
>
> "For MNIST vs greyscale SVHN, I see in fig 4 that NUQ does the same as the stupid entropy, which is interesting to observe and suggests that in this case it is easy to detect these OOD examples."
>
>
> You probably mean figure 3, as on figure 4, NUQ does almost the same as the perfect rule, while entropy does not differ much from random. On fig. 3 it is a simpler task, we extended the caption to make it more clear.
>
>
> --------------
>
> "I also notice that none of the more 'serious' competitive baselines (like Ensembles and DDU) were included in the evaluation. Why?"
>
>
> The evaluation with MNIST was not supposed to be a benchmark but rather an illustration as a sanity check that method generally works. We added these plots with extra baselines (DDU and ensembles) in the appendix A.3.
>
>
>
>
> It is pretty costly to train ensembles on CIFAR-100 and ImageNet. We did not tune this particular hyperparameter (the number of models in the ensemble) as we supposed that "the more, the better." Ultimately, we had 5 models. Here we want to emphasize that it is not practical to train "many models in the ensemble" as it is rarely applicable on practice.
> For a MC dropout, we grid-searched the optimal dropout rate on a misclassification detection on validation set. We should note, that a resnet canonically includes batch-norm, not dropout as a regularization. So we consider between last-layer dropout resnet50 and wide-resnet 28-10, but with changing the backbone architecture, we would need to re-train all other methods to compare in a meaningful way.
>
> -----------
>
> "How accurate is the proposed method in terms of estimating the TRUE epistemic uncertainty? In cases where the generative distribution is known, this should be easy to compute."
>
>
> Thank you for the suggestion. It is indeed an interesting experiment to check. We added corresponding results in the Appendix A.7. We see that NUQ allows to capture epistemic and aleatoric uncertainty very well.
>
>
> -----------
>
> "The proposed approach is suspicious on theoretical grounds, in that it relies essentially on the ratio of some estimation of training uncertainty and density in representation-space in order to estimate epistemic uncertainty."
>
>
> We do not consider it suspicious. It relies on the asymptotic behavior of classical Nadaraya-Watson estimator. The distinction between uncertainties is made based on proper derivations and allows us to distinugish these uncertainties in the fundamental way.
>
> -----------
>
> "The experimental results are also worrisome because it does not look like appropriate efforts were made to tune the hyper-parameters of baseline methods."
>
>
> We are sad to hear it and do not agree.  For the strongest competitor, DDU, we literally used the code authors released. For other methods we performed the full tuning of hyperparameters. Importantly, our results well correspond to ones in DDU paper.

---

> > ### Comment · Reviewer_CWxx · 2021-11-24
> > **Still not convinced**
> >
> > Thanks for the answers and the improvements to the paper. I am slightly raising my score, but it is still below the acceptance bar. See below.
> >
> > > "The proposed approach is suspicious on theoretical grounds, in that it relies essentially on the ratio of some estimation of training  uncertainty and density in representation-space in order to estimate epistemic uncertainty."
> >
> > > We do not consider it suspicious. It relies on the asymptotic behavior of classical Nadaraya-Watson estimator.
> >
> > Precisely what concerns me! In the asymptotic regime, the epistemic uncertainty goes to 0. Only considering training performance is completely oblivious to possible overfitting.
> >
> > > "Using the logits as input to the kernel machinery may be problematic, as highlighted for example by Postels et al 2021 (arXiv:2107.00649)"
> >
> > > The mentioned paper is indeed interesting, and we did not see it before your comment. It is interesting to investigate the effect  mentioned in the paper, especially considering how good deterministic neural nets work in uncertainty prediction. However, we should note that penultimate level features can still have directions close to invariance
> >
> > I am not saying that looking at the penultimate layer solves the problem, since it comes from dimensionality reduction (unless you have a wide top hidden layer compared with the input and no bottleneck in between). Hence, whether you apply your method on the output or the top hidden layer may still suffer from the problem raised by Postels et al 2021.
> >
> > > It is pretty costly to train ensembles on CIFAR-100 and ImageNet.
> >
> > Not a very convincing argument, especially for CIFAR-100. A 5-net ensemble is only 5x more expensive to train, which is very much feasible.

---

> > > ### Author Response · Authors · 2021-11-29
> > > **Comments on the theory part and additional experiments**
> > >
> > > With respect to the usage of asymptotic Gaussian approximation we have a certain misunderstanding, we believe. Of course, in the limit of an infinitely large sample size, epistemic uncertainty tends to zero. However, we don’t suggest considering infinitely large sample size. We took advantage of the fact that the Nadaraya-Watson estimator constructs prediction by summing up the output values for nearby points from the training sample with some weights. With the increase of sample size and assuming proper choice of bandwidth such sums will have the distribution tending to the Gaussian one (essentially some variant of Central Limit Theorem). We use the corresponding Normal approximation (and its variance in particular) for the fixed sample size.
> > >
> > > Exactly the same idea is used, for example, when one constructs confidence intervals for the parameters in statistical models with a finite number of parameters. Such intervals are not exact but with an increase of the sample size, they become closer and closer to the valid ones. In practice, already for very moderate sample size, such Gaussian approximation is accurate (for hundreds and sometimes even dozens of samples). Efficiently, such intervals are fully based on the training samples and don’t assume anything about the underlying probabilistic model except for certain smoothness constraints.
> > >
> > > In our case, the situation is the same. We might not have the perfect Gaussian approximation, but ultimately we just need it with a certain accuracy. The corresponding variance of NW prediction gives us an approximation of the epistemic uncertainty for the given sample size. The usefulness of the obtained estimator is then demonstrated in the experiments.
> > >
> > > Regarding ensembling, we’ve conducted an additional study with varying ensemble sizes for CIFAR-100 and ImageNet. For CIFAR-100 we considered ensembles of size 2,3,5,7 and 10 models. The corresponding OOD detection performance is:
> > > 1) SVHN 82.3+-1.3, 82.4+-0.7, 82.9+-0.9, 82.7+-0.7, 82.6+-0.5
> > > 2) LSUN 85.1+-0.6, 85.9+-0.6, 86.5+-0.8, 87.1+-0.6,  87.1+-0.6
> > > 3) SMOOTH 83.7+-6.5, 83.4+-3.2, 83.7+-1.2, 83.3+-1.5, 83.2+-1.6
> > >
> > > We observe the expected behavior of diminishing returns with the increase of the ensemble size. For SVHN we even don’t see an increase of the mean value with only variance decreasing. So, we can expect that the further increase in the ensemble won’t allow the ensemble to achieve the OOD detection performance comparable with NUQ/DDU on CIFAR-100.
> > >
> > > We observed similar behavior for Imagenet, where we have considered ensembles of sizes 2, 3, 5, 7, 9, and obtained the following results:
> > > 1) Imagenet-R 84.7, 85.3, 85.8, 86, 86.1
> > > 2) Imagenet-O 49.8, 50.8, 51.9, 52.1, 52.6
> > >
> > > Again we see the improvement with the increasing number of models but we can’t expect that the ensemble will reach the quality of NUQ for any reasonable number of models. Note that we have a typo in the table for Ensembles’ result on Imagenet-R in the text of the paper. The correct number is 85.8 (instead of 84.4 in the main text).
> > >
> > > Please also note that we performed additional experiments and compared NUQ with DUQ and SNGP, see the comments to other reviewers above.

---

### Official Review · Reviewer_S6bt · 2021-11-01

**Correctness:** 3
**Technical Novelty And Significance:** 2
**Empirical Novelty And Significance:** 2
**Recommendation:** 5
**Confidence:** 4

**Main Review:**



Reasons for score:

Although the idea is interesting, my main concern is about the significance of the proposed method. See my detailed comments below.

---

Pros:

- The paper uses the excess risk and Bayes risk to quantify the epistemic uncertainty and the aleatoric uncertainty, which is interesting.
- Practical method for large datasets is described, which is very important for nonparametric methods.
- Substantial experimental results are provided on several different image datasets.

---

Concerns:

My first concern is about the significance of the proposed method.

1) The method is actually only nonparametric in the feature space of a neural network. Therefore, the feature extracting process, which involves a large number of parameters, is extremely important. Although the uncertainty is defined rigorously, it is not clear whether it is still informative in the feature space of an arbitrary neural network. Previous work (Van Amersfoort et al., 2020;2021, Liu et al., 2020) shows that a crucial difficulty of uncertainty estimation in deterministic neural networks is feature collapse, i.e., the network maps both data samples and OOD samples to a very narrow region in the feature space, resulting in unexpected/uncalibrated predictive uncertainty. It seems the proposed method can also suffer from the feature collapse issue, while no theoretical analysis can be found in the paper (except that the issue is alleviated using spectral normalization, which is also proposed in previous work).

2) The authors claim that the proposed method complements recent works that suffer from a lack of a principled method to quantify the uncertainty. However, there are several other methods that can produce principled uncertainty. For example, being Bayesian in the last layer (Kristiadi et al., 2020), which is equivalent to using a Bayesian linear/logistic regression in the last layer, or using a Gaussian process in the feature space (Liu et al., 2020), which is also kernel-based and nonparametric. Given the fact that these methods can also distinguish epistemic uncertainty and aleatoric uncertainty, what are the advantages of the proposed method compared with these methods?

3) I also have a concern about efficiency. The authors described several existing methods as *still require setting additional hyperparameters and/or increased computational cost*. However, the proposed method requires extra GMM modeling, nearest neighbor search, and kernel bandwidth selection. I suggest the authors provide a detailed discussion and justification about the efficiency.


My second concern is about the experiments. As mentioned above, I think more methods should be compared, such as Kristiadi et al., 2020, DUQ and SNGP. The author has explained that the compared methods do not require significant modifications to model architecture and training procedures. However, given the fact that SN regularization is also adopted, the training procedure in the proposed method is actually modified and therefore the mentioned methods seem like fair competitors.

---

Reference:
Agustinus Kristiadi et al., 2020, Being Bayesian, Even Just a Bit, Fixes Overconfidence in ReLU Networks




**Summary Of The Paper:**

This paper proposes a nonparametric uncertainty quantification method for deep neural networks. Using a kernel-based estimator of the conditional density (i.e., the predictive distribution $p(y|x)$) in the feature space of a pre-trained neural network, the epistemic and the aleatoric uncertainty can be obtained separately from an upper bound of the excess risk and Bayes risk. For practical feasibility, several subproblems are approached with existing methods, such as kernel density estimation/GMM for $p(x)$, Improved Sheather-Jones algorithm for kernel bandwidth selection, HMSW for fast nearest neighbors retrieval. Finally, good OOD detection performance is demonstrated on several image datasets including MNIST, CIFAR, and ImageNet.

**Summary Of The Review:**

In summary, I raised several concerns about the theoretical and empirical significance of the proposed method, which currently prevents me from rating this paper higher. I will consider raising my evaluation if the authors address my concerns or point out my misunderstandings.

---

> ### Author Response · Authors · 2021-11-23
> **Good comments**
>
>
> Dear colleague, we would like to thank you for your thorough review! Please, see below the answers to your questions and concerns. Please, note that we didn’t perform any major updates to the main text and the answers for your particular concerns were put to Appendix of the paper. We do it for convenience of the review process and will update the main text accordingly if the paper is accepted.
>
> ------------
>
> "It seems the proposed method can also suffer from the feature collapse issue, while no theoretical analysis can be found in the paper (except that the issue is alleviated using spectral normalization, which is also proposed in previous work)."
>
> Indeed, our method is built on top of logits (or other embeddings) produced by a neural network. This means that our approach is sensitive to them. While we refer to the previous works on certain inductive biases (ResNet + Spectral Normalization), we surprisingly discovered that even without spectral normalization, our approach could capture uncertainties relatively well. It allows the method to be used on any network. We have a table in the appendix (Table 6), which shows this observation. Thus, in principle, one can apply NUQ to any model without retraining. Of course, if you can retrain it with spectral normalization, it would be better. We also should note that spectral normalization is ultimately not the guarantee as due to  successive application of many layers the Lipshitz constants may still become very small for the lower one and very large for the upper one. It is the limitation of the considered family of approaches.
>
>
> ------------
>
> "Given the fact that these methods can also distinguish epistemic uncertainty and aleatoric uncertainty, what are the advantages of the proposed method compared with these methods?"
>
>
> We were probably a bit hard on formulation, however we stress that in both mentioned papers no attempt to separate the types of uncertainty is done. It is possible to distinguish aleatoric and epistemic uncertainty for these methods but it will, to the best of our understanding, require sampling to perform sampling to estimate not only the mean (aleatoric) probabilities but also, for example, variances corresponding to epistemic uncertainty. Such a sampling might be costly compared to single pass over the data needed in for our method. We also think that relation of epistemic uncertainty to expected excess risk is interesting and will lead to future rigorous theoretical results on ood estimation. On top of (Kristiadi et al., 2020) and  (Liu et al., 2020), the majority of other "deterministic" uncertainty methods, are rather heuristical.
>
>
> ------------
>
> "... the proposed method requires extra GMM modeling, nearest neighbor search, and kernel bandwidth selection. I suggest the authors provide a detailed discussion and justification about the efficiency."
>
>
> Indeed, our approach requires tuning of some hyperparameters. But the cost of their computation is negligible. Exponential kernels are decaying rapidly, so there is no need to capture a lot of neighbors. We noticed that 20 neighbors are usually more than enough. The most expensive part is bandwidth selection. But here, we stress that it could be implemented so that you don't need to recalculate all the distances one more time. So the complexity of bandwidth selection is number_of_folds times HNSW search complexity. Most importantly, it is preprocessing step. So, at inference stage we work with fixed bandwidth.
>
> ------------
>
> "I think more methods should be compared, such as Kristiadi et al., 2020, DUQ and SNGP. The author has explained that the compared methods do not require significant modifications to model architecture and training procedures. However, given the fact that SN regularization is also adopted, the training procedure in the proposed method is actually modified and therefore the mentioned methods seem like fair competitors."
>
>
> Spectral normalization in modern frameworks does not lead to the actual change of training procedure. Indeed, each matrix of parameters is locally optimized to ensure Lipschitz, but the ultimate objective still stays the same. Moreover, our experiments (see Table 6 in the appendix) show that spectral normalization is essential but not a decisive component.
>
> For other methods the changes are more substantial. We also tried to implement SNGP and DUE (see the answer to Reviewer 1), but their peculiarities didn’t allow us to perform the testing in the timeframe given for rebuttal. Nevertheless, we will do all the efforts to perform comparison with mentioned method and add it to the camera-ready version of the paper (if accepted).

---

> > ### Author Response · Authors · 2021-11-29
> > **Additional experiments with DUQ and SNGP have been performed**
> >
> > Dear colleague, we additionally performed a series of experiments with SNGP and DUQ methods.
> >
> > To benchmark the DUQ on ImageNet we had to make some modifications to the original paper code. The original paper proposes to train DUQ end-to-end with gradient penalty, but we failed to make it converge on larger networks & datasets (ResNet-50 for Imagenet/CIFAR-100 for our experiments vs ResNet-18 for CIFAR-10 in the original paper) even after significant time spent on hyper-parameter search. Instead, we train the RBF head for a pre-trained backbone obtained from a spectrally normalized model.
> >
> > The results are:
> > 1) On CIFAR-100: SVHN 83.6+-4.0, LSUN 87.2 +-2.1, Smooth 83.8+-4.5
> > 2) On Imagenet: Imagenet-R 57.4, Imagenet-O 67.3
> >
> > We observe that the model beats the ensembles in all the cases except Imagenet-R, but it is inferior to DDU/NUQ. We conjecture that we
> >
> > Also, we performed a large-scale study of the SNGP method on Imagenet based on the available online implementation of SNGP by the authors of the original paper (though no public benchmarks of SNGP on Imagenet are available). We obtained the following results:
> > 1) Imagenet-R - 69.9
> > 2) Imagenet-O - 52.5
> > The results are inferior to the ones by NUQ and even DDU. We conjecture that the problem might be with the Random Fourier Features layer, which is quite different from the standard layers used in modern DNNs and significantly influences the convergence of the method.  It definitely requires careful treatment to make the model work well. Making a robust RFF-based GP method for large-scale image data is an interesting direction for future research.

---

> > > ### Comment · Reviewer_S6bt · 2021-11-29
> > > **Comments to Authors' Response**
> > >
> > > Thank you for your response and additional experimental results!
> > >
> > > ---
> > >
> > > > Indeed, our approach requires tuning of some hyperparameters. But the cost of their computation is negligible.
> > >
> > > OK.
> > >
> > > > We were probably a bit hard on formulation, however we stress that in both mentioned papers no attempt to separate the types of uncertainty is done. It is possible to distinguish aleatoric and epistemic uncertainty for these methods but it will, to the best of our understanding, require sampling to perform sampling to estimate not only the mean (aleatoric) probabilities but also, for example, variances corresponding to epistemic uncertainty.
> > >
> > > This is not very convincing. In Bayesian formulation, usually $p(f|D)$ represent epistemic uncertainty and $p(y|f)$ represent aleatoric uncertainty, which is quite common. And in the feature space, both linear models or GP (with variational approximation) can provide analytical predictive distributions, thus no sampling procedure is needed.
> > >
> > > ---
> > >
> > > > We observe that the model beats the ensembles in all the cases except Imagenet-R, but it is inferior to DDU/NUQ. We conjecture that we
> > >
> > > The author seems to have lost some explanations. What is the conjecture/conclusion of the results?

---

> > > > ### Author Response · Authors · 2021-11-29
> > > > **Addressing comments on well-gorunded Bayesian approaches to uncertainty estimation**
> > > >
> > > > Dear colleague, many thanks for your comments.
> > > >
> > > > Regarding the Bayesian part, we fully agree on the overall concept. However, in practice, SNGP uses the following approximation. As any Bayesian method, it targets estimating $\int p(y|f) p(f|D) d f$ which would require sampling in general. Instead, during training SNGP first computes $f^* = \int p(f|D) d f$ (this is done via predictive mean of GP) and then it uses $p(y | f^*)$ as an estimator for the integral. So, it is a certain type of Mean Field Approximation. At test time, sampling is done, see Algorithm 2 in https://arxiv.org/pdf/2006.10108.pdf. A similar situation is for Kristiadi et al. paper, where the LLLA method uses the same type of a trick, but also applies it at the inference stage.
> > > >
> > > > Overall, we admit that our method is indeed using another type of approximation (Gaussian one), so the methods have a comparable degree of rigor in terms of providing the theoretically grounded solution. We would like to thank you for this discussion and will correct our formulation in the camera-ready version of the paper (if accepted) discussing SNGP and LLLA as other well-grounded solutions. Nevertheless, our approach is quite different in nature and allows us to look at epistemic vs aleatoric uncertainties from a different perspective compared to Bayesian approaches. We also note that we now have an experimental comparison with SNGP which places our method better within recent approaches for uncertainty estimation and OOD detection.
> > > >
> > > > Regarding the missing explanation, it should be the follows "We conjecture that the DUQ model possibly might achieve the performance on par with DDU (at least there is nothing explicit that prevents the model from achieving it as the resulting description of the class (Gaussian) is the same as for DDU). However, its performance is naturally limited by an assumption on Gaussianity of each class which is too restrictive for complex data such as Imagenet."

---

### Official Review · Reviewer_EX9M · 2021-11-02

**Correctness:** 3
**Technical Novelty And Significance:** 2
**Empirical Novelty And Significance:** 2
**Recommendation:** 5
**Confidence:** 4

**Main Review:**

**Relevance**: The paper proposes a deterministic uncertainty estimation method for deep learning. Various other methods in that direction have been proposed recently (SNGP, DDU, DUE, etc) to avoid computing an expensive model average over multiple forward passes. The submission is therefore timely and relevant.

**Novelty**: My understanding is that the paper mostly combines results from the literature (the upper bound on the Bayes risk, the variance of the excess risk). This is of course a worthwhile contribution, but arguably at the lower end of the novelty spectrum. Prior work is clearly referenced.

**Clarity**: The high-level flow of the paper is clear and easy to follow. In some places I feel like the text could be expanded a bit to not only re-state results from prior works, but give the reader some intuition forwhy they are true and how they are derived. More importantly, I find the paper fairly unclear regarding the method that is ultimately implemented, section 2 ends fairly abruptly for my taste. I would suggest adding an algorithm or pseudo-code box to make it clear what exactly is implemented. This would also help to substantiate the statement that the paper presents a “fast and scalable method”.

**Empirical evaluation**: Overall the choice of problems and metrics is standard. My biggest concern is that the comparison to reasonable baselines is fairly lackluster. The small scale experiments (toy-classification and MNIST) only compare to deterministic networks which are well-known to not provide useful uncertainty estimates. So I find it hard to conclude much from these. An equivalent of Fig. 1 should be provided for DDU and the MNIST benchmarks from sections 4.2.1 and 4.2.2 should show results for ensembles and DDU (although I could imagine that the former won’t work very well out-of-the-box since MNIST is too simple, perhaps using something like FashionMNIST would be more interesting).

The ImageNet results look promising. It is a bit surprising, however, that NUQ is not able to outperform DDU on CIFAR100 (my impression is that usually if anything performance gains don’t translate from the smaller CIFAR datasets to ImageNet). The argument that the number of classes is too high for DDU is quite speculative. It seems to me like including another baseline method such as [SNGP](https://arxiv.org/abs/2006.10108) or [DUE](https://arxiv.org/abs/2102.11409) would solidify these results.

**Other notes/questions**:
* Just under eq (1): “In our experiments, we consider one dimensional kernel function”, but (2) has a power to dimensionality -- what exactly is happening here? Is the KDE fit jointly on all features or independent across dimensions?
* The statement in the 3rd point at the end of the introduction that recent work “suffered from a lack of a principled method to quantify the uncertainty” is extremely vague and should be substantiated (what do you mean by principled and how is prior work not principled?) or removed.
* “Thus, it might be beneficial to tune the bandwidth to optimize the quality of OOD detection if some set of OOD points is available at the training time”. I would ideally like to see a small ablation results on this, e.g. with the SVHN validation and test sets. It would be useful to know how much performance could be gained by tuning the bandwidth on a known OOD dataset.
* The captions of Fig 2 and 3 are too close to each other and should be spaced evenly with Fig 4.
* Broken link to supplement at the beginning of 4.2.1.
* The references are formatted extremely inconsistently. Please don’t just copy-paste bibtex entries from google scholar, but make capitalization, initials/full first names, venue names etc consistent.


**Summary Of The Paper:**

The paper proposes a post-hoc KDE-based method (NUQ) for single-pass uncertainty estimation. It reports improvements in robustness to data perturbation and OOD detection over deterministic networks on MNIST-based benchmarks, comparable performance to DDU (Mukhoti et al., 2021) on CIFAR100-based out-of-distribution detection and improvements on ImageNet.

**Summary Of The Review:**

Overall, this is an interesting, but not groundbreaking paper with a few issues around clarity and the empirical evaluation, so I am currently **leaning towards a reject**.

---

> ### Author Response · Authors · 2021-11-23
> **Good comments, answers are below**
>
> Dear colleague , we would like to thank you for your thorough review! Please, see below the answers to your questions and concerns. Please, note that we didn’t perform any major updates to the main text and the answers for your particular concerns were put to Appendix of the paper.
>
> While your understanding is correct, we do not agree on “the lower end of the novelty spectrum”. We indeed are basing on the recent results on uncertainty estimation for neural networks. However, we added some rigour to this stream of research by bringing classical results to derive our method. So, it is not simply the next iteration on top of what other colleagues did. Our paper instead opens a room for kernel-based methods to quantify uncertainty. We stress that epistemic uncertainty, in our case, has a proper statistical meaning (expected excess risk). In the majority of other "deterministic" uncertainty methods, it is rather heuristical. See also the experiments in Section A.6 which show the feasibility of the proposed method in the scenario with known uncertainties.
>
> We agree that adding a pseudo-code would simplify reading and enhance understanding. We currently added it to Appendix A.7 and are thinking of ways to incorporate it to the main text which is challenging due to space constraints.
>
> We agree that it might be beneficial to add some other methods to comparison. However,  we want to emphasize that the DDU method was shown to have superior performance over SNGP on CIFAR-100, see [Mukhoti2021]. We also should note that both these methods require more substantial changes to the training and inference procedures compared to NUQ, which requires nothing at all, but benefits from simple spectral normalization. In particular, we tried to make these methods work on ImageNet but failed in the time given for rebuttal (though it doesn’t look impossible).  From the application perspective, the predictive model stays the same for NUQ,while it changes for SNGP and DUE which might be important for many application areas where practitioners have their established models.
>
> Another possible issue is in potential problems with scalability for SNGP and DUE. All these papers only report results on CIFAR-100 and omit ImageNet, while we know that GP-based methods are not very scalable (even with all modern approaches to speed up them). Nevertheless, we will do all the efforts to perform comparison with SNGP and DUE and add it to the camera-ready version of the paper (if accepted).
>
> The main goal of this Figure 1 was to provide an intuition of how good NUQ works. It was not supposed to be a real benchmark for comparison between strong methods but rather an illustrative example. Nevertheless, we added a figure with extra baselines (DDU, ensemble) to Appendix A.3. We discuss in the text some benefits of NUQ in this comparison.
>
> We agree that the argument on the number of classes is indeed a bit speculative. We performed a visualization (see Appendix A.5) which shows that in fact the embedding structure is more complex for Imagenet which might be an explanation why DDU is not very good for Imagenet (a Gaussian per class is not enough).
>
> We indeed had certain inconsistencies in our notation for kernels. First of all, the resulting kernel is multidimensional: equation (1) shows how we build a multidimensional kernel from 1-dimensional. The bandwidth values can be equal for different dimensions or can be different. In the methodological part of the paper we put the same kernel with for all the dimensions for notational simplicity. In practice, ISJ bandwidth estimator gives us different values for each dimension. Additionally, in formula 2, the integral should have been in power d (fixed now).
>
> By this point, we refer to the previous papers on deterministic uncertainty quantification, mainly [Mukhoti2021]. They just use a marginal data density as a measure of epistemic uncertainty without any theoretical motivations. In principle, we could use any other statistic based on it. They discussed neither other choices nor why p(x) is superior. We show how we can obtain the estimate in a way directly related to the formal definitions of uncertainty.
>
> [Mukhoti2021] Mukhoti, J., Kirsch, A., van Amersfoort, J., Torr, P. H., & Gal, Y. (2021). Deterministic neural networks with appropriate inductive biases capture epistemic and aleatoric uncertainty. arXiv preprint arXiv:2102.11582.
>
> That’s another very good point, thank you. However, the ablation study with SVHN is not very meaningful as the method performs close to the optimum already. Instead, we performed an additional ablation study for CIFAR-100 with Smooth dataset being OOD and didn’t observe an improvement. We observe no benefit from considering OOD set, which might mean that our bandwidth selection procedure is already good enough. We will additionally perform a larger ablation study

---

> > ### Comment · Reviewer_EX9M · 2021-11-24
> > **Response**
> >
> > Thank you for your extensive response.
> >
> > Re novelty: by "lower end of the spectrum" I meant to express that the novelty is sufficient for acceptance (as indeed you are doing some work to develop the method), but not a strength of the paper.
> >
> > My main concern remains that in the ImageNet experiment, which I would argue is the key result in favor of this paper, it is simply not clear whether the proposed method works particularly well or DDU (as the only strong baseline besides ensembles) works particularly poorly. While the DDU paper may have reported improved performance over e.g. SNGP on CIFAR100, that does not imply that it also works better on ImageNet. It may well be the case that all recent methods on deterministic uncertainty for deep NNs struggle to scale, but that would have to be established with a little bit more empirical rigor than just a single baseline. I will therefore remain with my score.

---

> > > ### Author Response · Authors · 2021-11-29
> > > **Additional experiments with DUQ and SNGP have been performed**
> > >
> > > Many thanks for your comments. We additionally performed a series of experiments with SNGP and DUQ methods.
> > >
> > > To benchmark the DUQ on ImageNet we had to make some modifications to the original paper code. The original paper proposes to train DUQ end-to-end with gradient penalty, but we failed to make it converge on larger networks & datasets (ResNet-50 for Imagenet/CIFAR-100 for our experiments vs ResNet-18 for CIFAR-10 in the original paper) even after significant time spent on hyper-parameter search. Instead, we train the RBF head for a pre-trained backbone obtained from a spectrally normalized model.
> > >
> > > The results are:
> > > 1) On CIFAR-100: SVHN 83.6+-4.0, LSUN 87.2 +-2.1, Smooth 83.8+-4.5
> > > 2) On Imagenet: Imagenet-R 57.4, Imagenet-O 67.3
> > >
> > > We observe that the model beats the ensembles in all the cases except Imagenet-R, but it is inferior to DDU/NUQ.
> > >
> > > Also, we performed a large-scale study of the SNGP method on Imagenet based on the available online implementation of SNGP by the authors of the original paper (though no public benchmarks of SNGP on Imagenet are available). We obtained the following results:
> > > 1) Imagenet-R - 69.9
> > > 2) Imagenet-O - 52.5
> > > The results are inferior to the ones by NUQ and even DDU. We conjecture that the problem might be with the Random Fourier Features layer, which is quite different from the standard layers used in modern DNNs and significantly influences the convergence of the method.  It definitely requires careful treatment to make the model work well. Making a robust RFF-based GP method for large-scale image data is an interesting direction for future research.

---

### Decision · Program_Chairs · 2022-01-20

**Decision:**

Reject

**Comment:**

The paper proposes a simple approach to quantify uncertainty in "deterministic" neural networks, not unlike the works of SNGP, DDU, and DUE, where one only performs one forward pass rather than in an ensemble or Monte Carlo sample. In particular, they propose a kernel-based method on a network's logits to estimate uncertainty, obtaining data and model uncertainty estimates separately using a bound on Bayes risk.

While I agree with the relevance of the problem, there's a shared concern among reviewers across both technical novelty and experimental validation---particularly compared to prior work that can be difficult to understand the key distinguishing factor. I recommend the authors use the reviewers' feedback to enhance their preprint should they aim to submit to a later venue.